# The Role of Intracellular Potassium in Cell Quiescence, Proliferation, and Death

**DOI:** 10.3390/ijms25020884

**Published:** 2024-01-10

**Authors:** Irina I. Marakhova, Valentina E. Yurinskaya, Alisa P. Domnina

**Affiliations:** 1Department of Intracellular Signalling and Transport, Institute of Cytology of the Russian Academy of Sciences, Tikhoretsky Avenue 4, 194064 Saint-Petersburg, Russia; 2Department of Molecular Cell Physiology, Institute of Cytology of the Russian Academy of Sciences, Tikhoretsky Avenue 4, 194064 Saint-Petersburg, Russia

**Keywords:** potassium, cell volume, cell water, quiescence, cell cycle, proliferation, apoptosis, human mesenchymal stem cells, human blood lymphocytes

## Abstract

This brief review explores the role of intracellular K^+^ during the transition of cells from quiescence to proliferation and the induction of apoptosis. We focus on the relationship between intracellular K^+^ and the growth and proliferation rates of different cells, including transformed cells in culture as well as human quiescent T cells and mesenchymal stem cells, and analyze the concomitant changes in K^+^ and water content in both proliferating and apoptotic cells. Evidence is discussed indicating that during the initiation of cell proliferation and apoptosis changes in the K^+^ content in cells occur in parallel with changes in water content and therefore do not lead to significant changes in the intracellular K^+^ concentration. We conclude that K^+^, as a dominant intracellular ion, is involved in the regulation of cell volume during the transit from quiescence, and the content of K^+^ and water in dividing cells is higher than in quiescent or differentiated cells, which can be considered to be a hallmark of cell proliferation and transformation.

## 1. Introduction

The movement of ions, associated with the activity of ion pumps, channels and co-transporters, is involved in maintaining cellular stability under changing internal and external conditions. The main function of ion transporters is to mediate cell interaction with the environment and participate in cellular functioning. Monovalent ions are implicated in cell functions in many ways, from the classic mechanisms in maintaining membrane potential to the control of pH, Ca^2+^ homeostasis, cell volume, growth factor release, interaction with the extracellular matrix and so forth. Moreover, monovalent ion fluxes can modulate the homeostatic balance of animal cells by changing both the ion and water content, which can lead to changes in the concentration of intracellular ions, various signaling molecules, metabolites and macromolecules. Therefore, ion fluxes across the plasma membrane have a profound impact on cell viability and function. Failure of these ionic events can impair specific cell function and can also signal the cell to die.

Movement of the monovalent ions K^+^, Na^+^, and Cl^−^ has been described as a contributing factor to the regulation of cell proliferation. It has long been recognized that ion fluxes from altered activity of the Na/K pump and ion channels are necessary events in the rapid, receptor-mediated cellular response to growth factors [1,2,3,4,5]. At present, Na^+^ and Cl^−^ are known to influence intracellular pH, membrane potential and Ca^2+^ concentration in the cytoplasm and participate in the network of signaling events that control the activity of cell cycle regulatory proteins. Na^+^/H^+^ exchanger activity regulates G2/M progression by increasing pH_i_, which in turn regulates cyclin B1 expression and cdk2 activity [6,7,8]. Cellular Cl^−^ participates in cell membrane hyperpolarization during G1/S transition [9]. It is also noted that monovalent ions are involved in long-term cellular response during the transition from a resting state to proliferation and accompany cell transformation [10,11,12].

In this review, we limit our discussion to the role of K^+^ in controlling cell homeostasis during cellular growth, proliferation and programmed cell death. K^+^ permeates across the cell membrane through various ion channels and co-transporters and is the most abundant monovalent ion in the cell. A decline in the intracellular K^+^ concentration due to blocking of the Na/K pump slows down cell proliferation, arresting cells in the G1 phase of the cell cycle [13,14,15]. The role of K^+^ in the proliferative activity of cells is the subject of many review articles, but they are focused on K^+^ transport systems (mainly K^+^ channels) and their structure, properties and regulation in target strategies [16,17,18,19]. This review article discusses the mechanism by which K^+^ is involved in the regulation of proliferation and apoptosis. We focus on K^+^ content and transmembrane K^+^ fluxes in cultured animal cells to demonstrate the relationship between K^+^ transport and cellular growth, and we discuss how K^+^ can contribute to fundamental changes in cell state leading to the transition from a quiescent state to proliferation, senescence or apoptosis.

## 2. Changes in Monovalent Ion Homeostasis during Cell Culture

Basic information about the participation of ions in the regulation of cell growth, proliferation and death was obtained from cells in culture. When maintained as a monolayer culture, animal proliferating cells are characterized by peculiar changes in ion homeostasis that accompany cell cultured life and growth under dense cell conditions. In early studies of ionic changes in growing cultures of transformed cells with a high proliferation rate, it was noted that during the transition from a sparse to a dense cell layer, the intracellular content of K^+^ and Na^+^ and the transmembrane fluxes of these ions change in a particular way [20,21,22]. This observation was confirmed in studies with human endometrial mesenchymal stem cells (eMSC), which were used as a model of normal proliferating cells [23]. In the first days after cell seeding, in a sparse cell culture, the K^+^ content increases, and the Na^+^ content decreases mainly due to an increase in ion fluxes mediated by the Na/K pump. With exponential growth of a cell culture, the intracellular K^+^ content remains high, and then in a dense monolayer, the K^+^ content and K^+^ influx gradually decrease (Figure 1A,B). In proliferating cells of various origins, both normal and transformed, the K^+^ content, estimated as the amount of K^+^ per cell protein mass (K_i_/g protein) ranges from 0.9–1.0 mmol/g (exponentially growing culture) to 0.6–0.7 mmol/g (dense proliferating culture) (Table 1). Notably, under the same conditions, the cell Na^+^ content does not change; it remains low until a confluent cell monolayer of cells is reached. Furthermore, in dense, confluent cultures, K^+^ content in cells is maintained above the physiological level of internal K^+^ for cell survival (approximately 0.5 mmol/g) [13,14]. It should be emphasized here that in the reviewed studies, the data on cell K^+^ are presented as analytically measured ion content per protein mass in the cell (K_i_/g protein). In biology, such an assessment of the content of ions or other components in a cell is widely used. There are considerable difficulties in estimating intracellular ion concentrations (i.e., cell ion content related to cell water content) due to difficulties measuring the cell water or cell volume of attached and asynchronously growing cells in cultures. In addition, for K^+^, the calculation of intracellular content per protein mass in the cell (K_i_/g protein) is of particular interest. As the main intracellular cation, K^+^ can participate in the regulation of cell volume and intracellular water content; therefore, a change in the K_i_/g protein ratio for growing cells that are increasing in size may indicate a change in intracellular water content [24].

During culture growth, intracellular K^+^ content varies depending on both the density and the age of the cell culture. This feature of ion homeostasis was most clearly shown in the study of eMSCs, which are usually maintained in culture for a long time (Figure 1D). In both sparse and confluent cultures, in which cells were seeded on the same day but with different cell densities and the intracellular ion contents were assayed simultaneously, on the third day after seeding, the K^+^ content was lower in a culture with higher density than in a sparse culture (Figure 1C). In addition, the intracellular K^+^ content varies depending on the age of eMSCs, such that at the same monolayer density in early-passage cultures (up to 6–7 passages), the K^+^ content in cells is higher than in late-passage cultures. In contrast to K^+^ differences, no significant differences in intracellular Na^+^ content were observed between young and old eMSCs.

As cell cultures grow, the decrease in K^+^ content correlates with the accumulation of cells in the G1 phase of the cell cycle, both in a confluent monolayer and in late-passage “old” cultures (Figure 1E,F). Growth-related changes in K^+^ content occur in both transformed and normal cycling cells and reflect changes in the proliferative activity of the cell population. Altogether, these studies indicate that a higher intracellular K^+^ content accompanies successful cell proliferation, and the ratio of intracellular K^+^ content to protein mass in the cell (K_i_/g protein) can be used as a physiological marker of cell proliferative activity.

## 3. K^+^ and Proliferation Arrest: Lower Cell K^+^ Content Accompanies Stress-Induced Senescence and Transit from Monolayer (2D) Conditions to Three-Dimensional (3D) Culture of Human Stem Cells

Senescence, which can be triggered in cells in response to various intrinsic and extrinsic stimuli as well as developmental signals, is a state of indefinite cell cycle arrest associated with aging, cancer, and age-related diseases [31,32,33]. Cellular senescence is commonly induced by inhibiting the cell cycle through DNA damage. Senescence is a dynamic process; it is progressive, multistep and ultimately irreversible, in contrast to a quiescent state, which is reversible. Growth arrest is a way to avoid replication of damaged or transformed cells and is sustained by the p16/Rb and p53/p21 signaling pathways. Progression from early (reversible) to full (irreversible) senescence implies extensive chromatin remodeling, mitochondrial alterations, increased autophagy and the production of a senescent secretome [34,35,36,37]. The accumulation of damage to DNA, proteins, lipids and carbohydrates is considered to be responsible for the functional decline occurring during cell senescence.

Senescent cells remain metabolically active for a long time. Specific data indicate changes in some parameters of ion homeostasis during senescence development. A higher concentration of intracellular Ca^2+^ has been observed in senescent cells compared with proliferating cells [38,39]. Fluorescent probes have revealed an increased content of both K^+^ and Na^+^ in the senescent human lung fibroblast IMR90 [40]. However, thorough estimations of ions in cells with the flame photometry technique have shown that stress-induced senescent human eMSCs have low Na^+^ and maintain a high K^+^/Na^+^ ratio typical for functionally active animal cells [25]. Indeed, short oxidative stress leads to a decrease in intracellular K^+^ content and an increase in Na^+^ content in eMSCs; however, within a day, in “early” stress-arrested cells, the ionic gradients and ion pump activity are restored (Figure 2). Later, during senescence progression, the Na^+^ content is slightly increased and the pump-mediated K^+^ transport is enhanced. The higher K^+^ influxes in “late” stressed senescent eMSCs are not associated with modulation of the Na/K pump properties or an increased number of ion pumps but are due to higher cell Na^+^ content in late senescence [25]. Stress-induced proliferation arrest does not affect the ouabain-resistant K^+^ fluxes associated with the activity of ion channels and ion co-transporters. During long-term culture, senescent cells maintain the high K^+^/Na^+^ ratio typical for functionally active animal cells.

The most significant observation regarding ions during the development of stress-induced senescence in eMSCs concerns the intracellular K^+^ content [25]. The senescence progression is accompanied by a gradual decrease in the intracellular K^+^ from 0.7 mmol/g protein (“early” senescent eMSCs) to 0.5–0.6 mmol/g protein (“late” senescent eMSCs) (Figure 2A). A peculiar feature of senescent eMSCs, namely a lower cell K^+^ content per g of cell protein (K_i_/g protein), is consistent with the above assumption that a decrease in this index indicates a decrease in the rate of cell proliferation.

The relationship between cellular K^+^ and cell proliferation rate is found in three-dimensional (3D) cell cultures. Different 3D cultures, which originally emerged as aggregates of tumor cells, are used for the cultivation of stem cells [41,42,43]. Culturing of cells in a 3D configuration is more appropriate in clinical trials; however, in these systems, the cell phenotype and cell behavior are altered [44,45]. In human eMSCs, significant modulations of ion homeostasis occur during the 2D/3D transition. Detached from a 2D culture, eMSCs retain high amounts of K^+^ but accumulate Na^+^ and have dissipated transmembrane ion gradients, while in compact scaffold-free 3D spheroids, eMSCs restore the lower Na^+^ content and the high K^+^/Na^+^ ratio [26]. In scaffold-free spheroids, eMSC are non-proliferating cells with an active Na/K pump and a high K/Na ratio, but with a lower K^+^ content than that of eMSCs in a 2D culture (Table 1). Non-proliferating eMSCs in spheroids are viable cells that begin cycling when seeded into a monolayer 2D culture. Notably, cell adhesion in 2D culture and the transition to proliferation are associated with an increase in the K^+^ content in eMSCs.

Changes in the ionic homeostasis of eMSCs during 2D/3D/2D transitions are associated with changes in both the cell–extracellular matrix and cell–cell contacts and are mediated by changes in the ion permeability of the cell membrane. The high intracellular Na^+^ content in suspended eMSCs can be explained by an increase in the Na^+^ permeability of the cell membrane due to the loss of cellular contacts with the matrix and actin disassembly [46]. The spheroids in hanging drops are a scaffold-free culture system in which cell contacts with the extracellular matrix are destroyed, but E-cadherin-mediated contacts among cells are enhanced [47,48]. The membrane-incorporated Na/K-ATPase promotes the formation of tight junctions among cells due to the adhesive properties of the β-subunit of the Na/K-ATPase pump: E-cadherin binds to the β-subunit, forming a complex for adhesion [49,50,51]. Consequently, cell–cell interactions during the formation of spheroids can help optimize the activity of the ion pump and normalize the ion homeostasis of the cells forming the 3D culture.

Human eMSCs in 3D spheroids have a lower K^+^ content (0.5–0.7 mmol/g) than eMSCs in a pre-confluent monolayer culture with a high rate of proliferation (0.9–1.0 mmol/g) (Table 1). In compact 3D spheroids, eMSCs are viable but non-cycling cells, such that a low cell K_i_/g protein ratio in 3D eMSCs is consistent with their loss of proliferative activity and confirms the general conclusion that a decrease in this index indicates a decreased proliferative status of the cell population.

## 4. The Mechanism of K^+^ Involvement in Cell Proliferation; Changes in Intracellular K^+^ and Water Content during the Transition of Human Lymphocytes from Quiescence to Proliferation

K^+^ is the main cation in the cytoplasm of most animal cells; it is a housekeeping element involved in not only the establishment of the electric potential differences across the cell membrane but also in the transmembrane transport of water and regulation of cell volume. Changes in K^+^ transport associated with the proliferative status of cell culture may reflect the involvement of K^+^ in cell transition from a resting state to proliferation [10,27]. Human peripheral blood lymphocytes (HBLs), which are quiescent cells, represent an excellent model for studying the events underlying the cell transition from quiescence to cell cycle and further continuous proliferation. In quiescent HBLs, antigen recognition with appropriate co-stimulation triggers an exit from a quiescent state (G0) and a transit through cell cycle (G1→S/G2/M) to proliferation.

In activated HBLs, in the first hours of antigenic stimulation, a decrease in K^+^ content occurs, followed by a long-term elevation of intracellular K^+^ during transition to DNA synthesis and cell division (Figure 3A). The initial decrease in K^+^ content (from 0.6 to 0.5 mmol/g) takes place simultaneously with an approximately twofold increase in the cell Na^+^ content (from 0.11 to 0.21 mmol/g) as well as increased K^+^ uptakes via Na/K pump, bumetanide-sensitive cotransport and ouabain-resistant K^+^ leakage [12]. At later stages, intracellular K^+^ increases to 0.8–0.9 mmol/g, while the Na^+^ content in cells remains low and unchanged. In activated HBLs, a long-term increase in cell K^+^ content is associated with IL-2-dependent progression, when small T cells are transformed into blasts (Figure 3B). Drugs specific to different steps of the G0/G1/S transit eliminate the long-term increase in both K^+^ content and K^+^ fluxes, thus indicating the intimate relations between delayed, sustained changes in K^+^ transport and cell transit from a quiescent state to proliferation [12,27]. Late long-term activation of the Na/K pump is due to increased expression of the Na/K-ATPase protein and mRNA [12,52]. Na/K-ATPase is implicated in the regulation of ion and osmotic balance in activated HBLs; it is conceivable that the enhanced pump should be involved in the genesis of cell growth response to mitogens, thus participating in cell cycle regulation at the physiological level.

Given significant changes in the intracellular K^+^ content estimated as K_i_/g cell protein, it is important to know whether in this case the intracellular K^+^ concentration (the K^+^ content in cell water) changes. A determination of the water content in HBLs by measuring cell density on the Percoll gradient suggests that activated HBLs are distributed mainly in two high-density regions: at 1.0645 g/mL and at 1.0695 g/mL (Figure 4A). Based on the buoyant density estimates, in HBLs stimulated to transit from quiescence to proliferation, cell water content rises, with the most significant increase (approximately 27%) during the first activation day (Figure 4E,F). The water content is also higher in continuously proliferating human leukemia Jurkat T cells than in resting HBLs (Figure 4E,F). Altogether, simultaneous determinations of cell K^+^ content via flame photometry and cell water content by measuring the cell buoyant density as well as cell volume using a Coulter counter show that in activated HBLs, K^+^ content and water content increase simultaneously (Figure 4). These data allow us to conclude that no significant differences in K^+^ concentration exist between quiescent and cycling cells. The following question then arises: What is the functional significance of the increase in cell K^+^ content during the transition from a quiescent state to proliferation? Is cell K^+^ essential for the initiation of cell proliferation in the context of cell volume regulation?

Cell transition from quiescence to proliferation is accompanied by an increase in cell size. Before division, the activated cell is growing, and cell volume is significantly increased. It has been shown that experimentally induced changes (decreases or increases) in cell volume inhibit culture growth and cell proliferation [53,54,55,56]. From the “pump–leak“ theory of monovalent ion distribution between animal cells and the medium, it follows that the amount of K^+^ in a cell essentially depends on the amount of “impermeant” (through cell membrane) anions ”sequestered” in the cell. It is the amount of these anions in combination with the Na/K pump that determines the water balance of the cell and the accumulation of K^+^ inside the cell [24,57,58,59,60]. In quiescent cells (such as human HBLs) that are stimulated to begin the division cycle, during the G0/G1/S progression, the increase in protein mass and volume is accompanied by an increase in the quantity of impermeable anions, which inevitably leads to an increase in the influx of water to restore the osmotic balance between the cell and the environment. Here, K^+^ as the major cellular osmolyte enters the cell so that an increase in the cell K^+^ content/protein mass ratio always correlates with an increase in the cell water content/protein mass ratio. Therefore, despite the absence of significant concentration differences, K^+^ may be important for the initiation of cell proliferation as an intracellular ion involved in the regulation of water content in cells, and a higher K^+^ content in proliferating cells indicates a higher hydration of cycling cells compared with quiescent and differentiated animal cells.

Cell hydration can significantly impact the biochemical life of the cell. Under physiological conditions, changes in cell volume and water content influence intracellular signaling, protein transport and gene expression [61,62,63,64]. Cellular dehydration may underlie cellular senescence [65]. A recent report on mouse hematopoietic stem cells suggested a causal link between cell size and stemness, whereby the enlargement of stem cells reduces their stem cell potential [66]. Cells tightly control their size and maintain a constant concentration of macromolecules in the cytoplasm [67,68]. Changes in the degree of cellular hydration will alter the intracellular crowding of macromolecules. To coordinate changes in cell volume, water dynamics and intracellular crowding, WNK kinases have been proposed: Being sensitive to volume changes, these kinases regulate the activity of ion transporters in the cell membrane and may also act as a sensor of physiological crowding [69,70,71]. Together, these studies indicate an important association between cell growth, cell hydration, and cell metabolism as a whole, which might be involved in the regulation of cell proliferation. This review highlights a link between intracellular K^+^ and cell proliferation and growth, which is mediated by regulation of cell volume and leads to a constancy of cellular K^+^ concentration.

## 5. Cellular K^+^ in the Induction of Apoptosis

Apoptosis, a programmed cell death, is a fundamental biological mechanism to remove unwanted, aged, defective or potentially harmful cells without affecting neighboring healthy cells. Loss of cell volume or cell shrinkage is a peculiar feature of this physiological cell death, along with chromatin condensation, internucleosomal DNA fragmentation, the formation of apoptotic bodies and the disintegration of mitochondria (reviews [72,73]). According to numerous studies, changes in the content of ions, primarily K^+^, play a key role in the progression of apoptosis. The movement of ions is an important part of the cell death process, followed by an apoptotic volume decrease (AVD) [74,75]. AVD is commonly considered to be a hallmark of apoptosis; however, whether a cell shrinks during apoptotic death appears to depend on both cell type and stimulus [76,77]. Nevertheless, with or without cell shrinkage, apoptosis is accompanied by a change in the content of K^+^ and Na^+^ in cells and a decrease in the K/Na ratio. Many researchers have observed a decrease in the concentration of K^+^ during apoptosis. With fluorescence probe flow cytometry analysis and inductively coupled plasma/mass spectrometry, loss of intracellular K^+^ was detected in Jurkat cells during anti-Fas-induced apoptosis and in dexamethasone-treated thymocytes [78,79,80,81]. It should be noted that flow cytometry data cannot always be interpreted unambiguously. A decrease in the fluorescence intensity of the ion probe can be associated not only with the release of ions from the cells, but also with an increase in the number of small cell particles formed during apoptosis (apoptotic bodies and cellular debris). Nonspecific leakage of fluorescent indicators across the plasma membrane in apoptotic cells can also be affected [77].

The key role of disruption of K^+^ homeostasis was demonstrated using fluorescent dyes in the induction of apoptosis in human glioblastoma LN229 cells [82]. Apoptotic changes in these cells were observed upon exposure to the specific Na/K-ATPase blocker ouabain (0.1–10 μM) and the K^+^ ionophore valinomycin. Apoptosis induction with ouabain in parallel with changes in ionic homeostasis was studied in human endometrial mesenchymal stem/stromal cells (eMSCs) in a 3D culture [26]. A decrease in the K^+^ content in eMSCs after inhibition of the Na/K pump led to a disruption in the mitochondrial potential and priming of cells to apoptosis.

A marked decrease in the K^+^ content and K/Na ratio in apoptotic cells of different species was shown with the aid of X-ray elemental microanalysis. The loss of K^+^ was detected in monocytes (macrophages), induced by 3 h of exposure to oxidized low-density lipoprotein [83] and in LNCaP prostate cancer cells induced into apoptosis via etoposide [84]. Another group of researchers using X-ray microanalysis performed a detailed study of changes in the ion content of human histiocytic lymphoma U937 cells induced into apoptosis with staurosporine, an intrinsic pathway activator, or UV irradiation [85,86,87,88]. They found a lower K^+^ and Cl^−^ content and a higher Na^+^ content in apoptotic cells compared with non-apoptotic control cells. The initial stages of apoptosis were characterized by a decrease in the level of K^+^ and Cl^−^ in all cell compartments [88]. The decrease in these elements preceded apoptotic changes in the nucleus and was the largest in mitochondria, where it occurred before the release of cytochrome *c.* Changes in the intracellular content of K^+^ and Na^+^ at the single-cell level were detected via X-ray spectral microanalysis during apoptosis of U937 cells induced with osmotic shock and etoposide [89,90]. An increased ratio of intracellular Na/K was detected for the majority of U937 cells entering apoptosis. With correlative light and cryo-scanning transmission electron microscopy, a stable decrease in K^+^ content was also observed in all cellular compartments of HeLa cells induced into apoptosis using actinomycin D [91]. Changes in the content of Na^+^ and Cl^−^ during apoptosis of HeLa cells occurred in two time steps.

Unlike X-ray microanalysis, in our studies, in parallel with the determination of the intracellular ion content during apoptosis, changes in water content in cells were determined, which made it possible to evaluate changes in the concentration of ions per volume of water in the cells. The ion content in the cells was determined with flame emission photometry, and the change in water content in the cells was estimated from the buoyant cell density on the Percoll gradient [29,76] (Figure 5). An increase in buoyant cell density, indicating cell shrinkage, was observed upon apoptosis of U937 cells induced with 1 μM of staurosporine (STS) for 4 h. This increase in density was accompanied by a decrease in K^+^ content (from 1.1 to 0.78 mmol/g protein), which exceeded the increase in Na^+^ content (from 0.30 to 0.34 mmol/g) and led to a significant decrease in total K^+^ and Na^+^ content in cells. In contrast to the effect of STS, treatment of cells with 50 μM of etoposide for 4 h or 0.8–8 μM of etoposide for 18–24 h induced apoptosis without triggering cell shrinkage. However, during etoposide-induced apoptosis of U937 cells, the intracellular K^+^ content and the K/Na ratio decreased, as in STS-treated cells. At the same time, the total K^+^ and Na^+^ content remained virtually the same due to the decrease in K^+^ content and the increase in Na^+^ content being near-identical. Since apoptotic water loss in STS-treated cells correlated with a decrease in total cellular K^+^ and Na^+^ content, no significant decrease in K^+^ concentration in cell water during apoptosis was observed. According to our data, the concentration of K^+^ in terms of cell water decreased by 8% during STS-induced apoptosis, and by 13.5% during etoposide-induced apoptosis [76]. The study of apoptosis of rat thymocytes induced by 1 µM of dexamethasone for 4–5.5 h or 50 µM of etoposide for 5 h confirm that the decrease in K^+^ content is one of the most significant factors leading to apoptotic cell shrinkage [29]. However, since the loss of intracellular K^+^ occurs in parallel with a similar loss of cell water, the cytosolic concentration of K^+^ in rat thymocytes is reduced by only about a third. Thus, simultaneous determinations of ion and water content show that with or without early AVD, the induction of apoptosis is always accompanied by a decrease in the K^+^ content in cells, which, however, does not lead to a significant decrease in the concentration of K^+^ in cells stimulated along the programmed pathway death.

It is generally assumed that a decrease in intracellular K^+^ concentration may play a key role in the regulation of apoptotic nucleases and caspases [79,92,93]. Another opinion is that a decrease in the concentration of intracellular K^+^ is not necessary for apoptosis [94,95]. Panayiotidis et al., in their review, discuss the question of how precisely K^+^ is linked to the apoptotic cascade in detail [96]. According to the authors, the role of intracellular K^+^ may relate not to the regulation of enzyme activity but to the control of activation and/or signaling events of the apoptotic process. Our data indicate that the K^+^ release during apoptosis induction is an underlying mechanism of early volume decrease (cell shrinkage). During this early apoptotic response, the change in intracellular K^+^ concentration is insignificant and is unlikely to affect the activity of apoptosis enzymes.

## 6. Concluding Remarks

Ion transport is an integral part of the regulation of a cell cycle. It is well established that ion channels and ion transporters are implicated in signaling in cells stimulated by growth-promoting factors. By contrast, the role of monovalent ions in long-term cellular responses to mitogen is less explored; meanwhile, ion fluxes across the plasma membrane participate in the homeostatic balance of the activated cells and can provide a specific intracellular ion context during a peculiar cellular response. In this regard, K^+^, as the main intracellular ion, is important for transition processes during the development of cell proliferative response as well as the induction of programmed cell death via apoptosis. The exit from a quiescent state to proliferation is accompanied by cell growth and an increase in cell volume, while the induction of apoptosis is accompanied by cell shrinkage. In both cases, cell volume regulatory mechanisms include water and K^+^ movement across the cell membrane, leading to a constancy of intracellular K^+^ concentration. The major conclusion from this review is that the water content in proliferating and transformed cells should be higher than in quiescent cells. In this regard, the water content of cancer cells is similar to that of embryonic tissue but consistently higher than that of normal cells of similar origin, whereas age-dependent loss of intracellular water and cell dehydration are concomitants of cellular senescence. Variations in the degree of cell hydration alter the crowding of macromolecules within the cell, thereby influencing the many regulatory events such as intracellular signaling, protein transport and epigenetic modifications. To determine if and how the water movement and hydration of living cells can be controlled, further research into ion transport pathways—both channels and co-transporters—is needed. This will extend our knowledge of the long-term regulatory mechanisms of cell proliferation, aging and death and help find reliable therapeutic approaches.

## Figures and Tables

**Figure 1 ijms-25-00884-f001:**
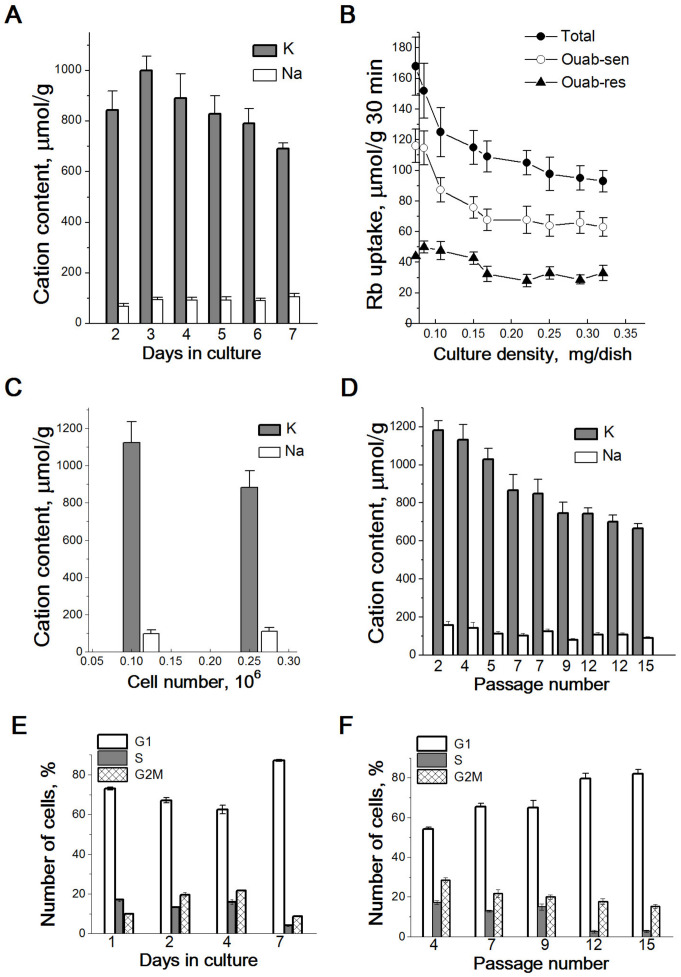
Proliferation-related changes in K^+^ and Na^+^ content and Rb^+^ influxes in growing cultures of eMSCs. Cell K^+^ and Na^+^ content (**A**) and Rb^+^ influxes (**B**) are dependent on the density of eMSCs culture. Total—total Rb^+^ uptake (as a mesure of K^+^ influx), Ouab-sen—ouabain-sensitive influx and Ouab-res.—ouabain-resistant leakage. (**C**) Increased culture density impact on intracellular cation content. Cells were seeded simultaneously at two densities (5 × 10^4^ and 15 × 10^4^ cells per 35 mm dish), and on the 3rd day, the intracellular cations were estimated. (**D**) Cell K^+^ content is decreased in the late-passage eMSCs. (**E**,**F**) The proliferation rate is decreased in high-density (**E**) and in late-passage (**F**) cultures of eMSCs. FACS analysis was performed on growing eMSCs within 7 days in culture (**E**) or with eMSCs of different passages on the 3rd day after plating (**F**). Data of 3–5 independent experiments are presented as means ± SD of 2–3 cell cultures on the day of experiment according to [23].

**Figure 2 ijms-25-00884-f002:**
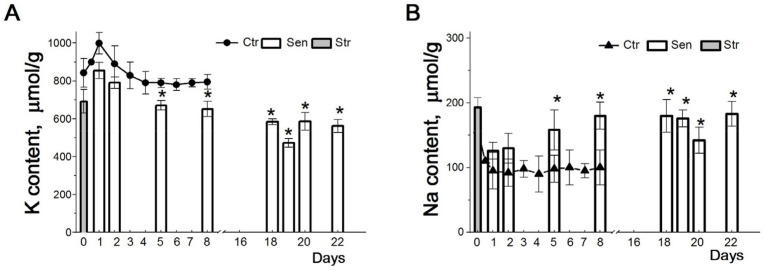
Changes in K^+^ and Na^+^ contents during stress-induced senescence in human eMSCs. Cells were seeded in 3-cm dishes (10 × 10^4^ cells per dish), and on the second day after seeding, some cultures were subjected to 200 μM H_2_O_2_ for 1 h, followed by H_2_O_2_ replacement and cell cultivation under normal conditions. The content of cellular K^+^ ((**A**), circles) and Na^+^ ((**B**), triangles up) in cycling MSCs during culture growth. The content of K^+^ ((**A**), columns) and Na^+^ ((**B**), columns) in cells during senescence progression. Grey columns represent cellular content of K^+^ (**A**) and Na^+^ (**B**) after treatment of cells with 200 μM H_2_O_2_. For the first 8 days, data are shown as means ± SD for six independent experiments performed in triplicate; significant difference between stress-induced and control proliferating cells (Ctr) was calculated using one-way ANOVA with Tukey’s post hoc tests, * *p* < 0.05. For late senescent cells (18–22 days), data are shown as means ± SD (*n* = 3) according to [25].

**Figure 3 ijms-25-00884-f003:**
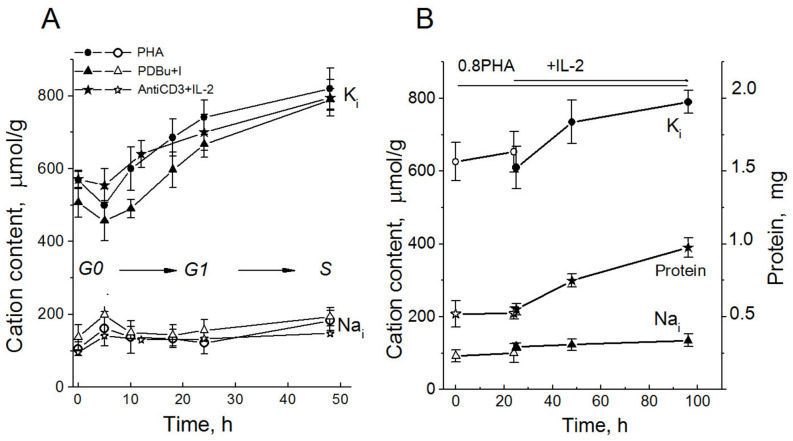
Cellular K^+^ and Na^+^ contents in human HBLs activated to transition from quiescence to proliferation. (**A**) Changes in the cellular K^+^ (K_i_) and Na^+^ (Na_i_) contents in human HBL stimulated by PHA (10 µg/mL), or PDBu (10 nM) with ionomycin (I, 500 nM) or anti-CD3 (3.5 µg/mL) with IL-2 (100 U/mL). The contents of cellular K^+^ (dark symbols) and Na^+^ (light symbols) were analyzed at definite time points via flame emission photometry. (**B**) IL-2 induces a long-term increase in cellular K^+^ and cellular protein content in competent HBL. Isolated HBL were incubated with non-mitogenic PHA (0.8 µg/mL) for 20 h, then IL-2 (100 U/mL) was introduced into cell culture. Data are means ± SD of 5 experiments performed triplicate. According to [27].

**Figure 4 ijms-25-00884-f004:**
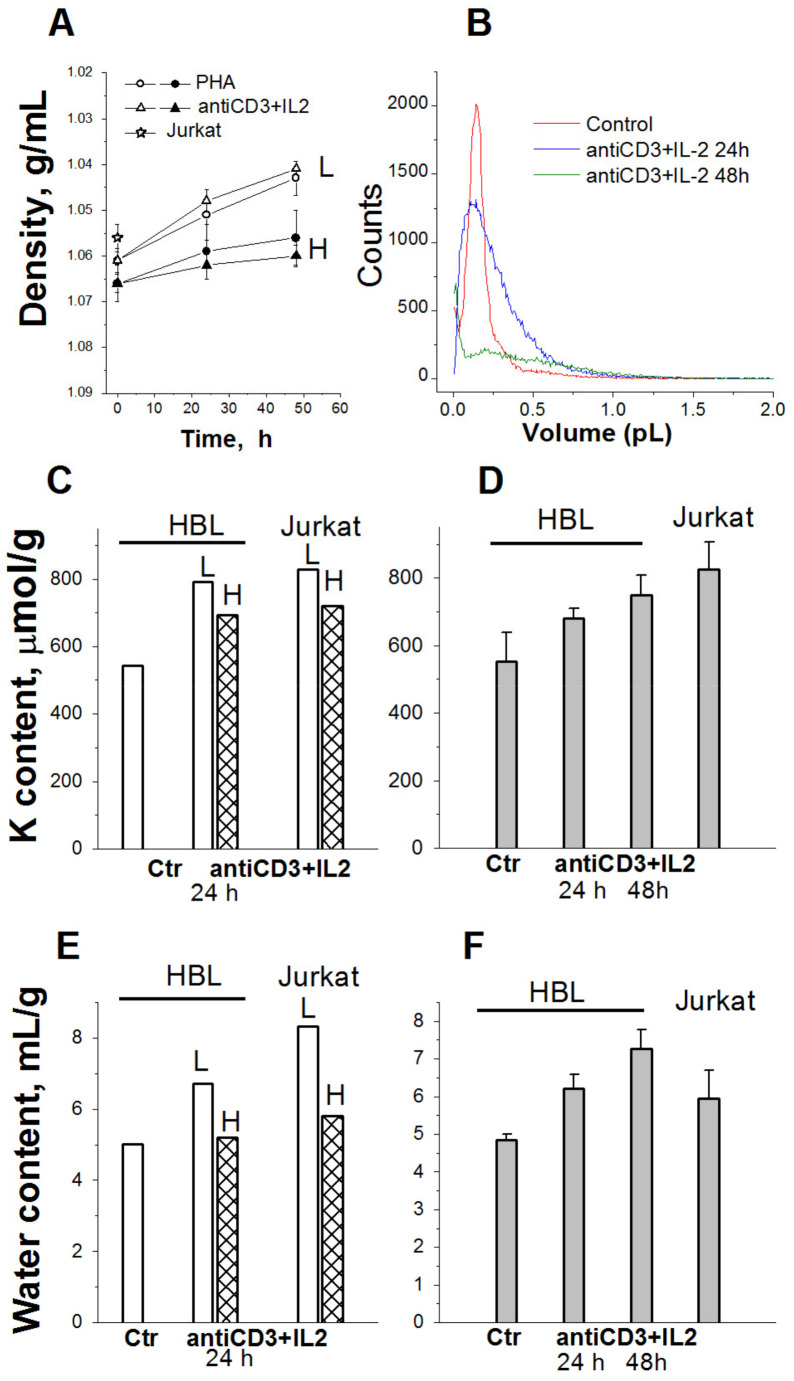
Cellular K^+^ and water content in HBLS with different buoyant densities compared to unfractionated activated HBLS and cycling T cells Jurkat. (**A**) Cell density decreases in HBLS stimulated from quiescence to proliferation. L—light and H—heavy cell populations. (**B**) Representative cell volume distributions obtained using a Scepter Counter for HBL, resting and stimulated VIA (anti-CD3 + IL-2) mixture for 24 and 48 h. Control—resting HBL. (**C**,**E**) Cellular K^+^ (**C**) and water content (**E**) in light (L) and heavy (H) cell populations stimulated with anti-CD3 with IL-2 and in proliferating T cells Jurkat. (**D**,**F**) Cellular K^+^ (**D**) and water content (**F**) in total unfractionated HBLs stimulated with anti-CD3 with IL-2 and in proliferating T cells Jurkat. One representative experiment performed in triplicate. Data are means ± SD. According to [27].

**Figure 5 ijms-25-00884-f005:**
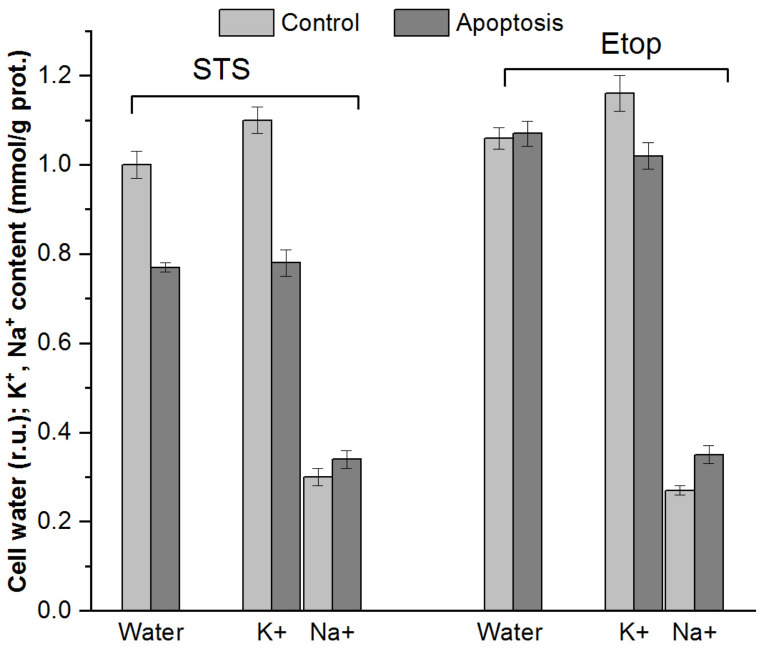
Changes in water and K^+^ and Na^+^ content in U937 cells induced to apoptosis by treatment with 1 µM of staurosporine (STS) or 50 µM of etoposide (Etop) for 4 h. According to [76].

**Table 1 ijms-25-00884-t001:** Variations in the content of K^+^ and Na^+^ in animal cells at quiescence, during proliferation and senescence.

Cell Type	Ion Content, mmol/g Protein			References
	At a cell culture density, ×10^4^ cells/cm^2^			
**Cells of permanent lines in 2D culture**	1–5	8–15	5–15	
	K^+^		Na^+^	
Mouse fibroblasts Swiss 3T3	0.8–0.9	0.65	0.2	[4,20]
Chicken embryo fibroblasts CEF	0.8	-	-	[21]
Chinese hamster ovary cells CHO-K1	0.8–0.9	0.6	0.2–0.3	[22]
Mouse fibroblasts L	0.9–1.0	0.6–0.7	0.15–0.2	[20]
Mouse fibroblasts L, serum-free	0.86–0.9	0.7	0.15–0.3	[20]
**Human stem cells**				
Endometrium mesenchymal stem cells, eMSCs	0.9–1.0	0.5–0.6	0.1–0.15	[23]
Senescent eMSCs	0.5–0.7		0.1–0.2	[25]
eMSCs in 3D culture	0.5–0.7		0.2	[26]
**Cells in suspension**	At a cell culture density (0.5–1.0) × 10^6^ cells/mL			
	K^+^		Na^+^	
Human blood lymphocytes				[10,27]
Quiescent G0	0.6		0.1–0.12	
Competent G0/G1	0.6		0.12	
Cycling G1/S	0.8–0.9		0.12–0.15	
Human leukemia T cells Jurkat	0.8–0.9		0.15–0.2	[27]
Human histiocytic lymphoma cells U937	0.8–1.1		0.27–0.30	[28]
Human erythroleukemia cells K562 *	0.9–1.1		0.1–0.15	
Rat thymocytes	0.95–0.98		0.17–0.25	[29]
Human red blood cells, RBC **	0.29–0.31		0.02–0.03	[30]

* Our unpublished data on K562 cells. ** Data on red blood cells are given in relation to the content of cellular hemoglobin [30].

## Data Availability

All data generated or analyze during this study are included within the manuscript.

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
