# Peer review of "The Role of Intracellular Potassium in Cell Quiescence, Proliferation, and Death"

_ijms, 2024, doi:10.3390/ijms25020884_

Round 1

Reviewer 1 Report

Comments and Suggestions for Authors

The review article entitled The role of intracellular potassium in cell quiescence, proliferation and death" explores the role of K+ in maintaining cellular homeostasis during the transition of cells from quiescence to proliferation and the induction of apoptosis. They focus on the  relationship between intracellular K+ , growth and proliferation rate of different cells including transformed cells in culture as well as human quiescent T cells and mesenchymal stem cells and analyzed the concomitant changes in K+ and water content in both proliferating and apoptotic cells. I think it represents a mini review in its current form and almost all sections need to be improved and elaborated extensively. The review article does not worth publishing as more extensive ones are already published in the literature such as:

https://link.springer.com/article/10.1007/s00726-013-1536-7

https://www.nature.com/articles/4400296

https://www.nature.com/articles/s41598-019-52571-1

https://www.nature.com/articles/s41598-018-36922-y

https://royalsocietypublishing.org/doi/full/10.1098/rstb.2013.0094

Comments on the Quality of English Language

Moderate editing of English language required

Author Response

Thank you for your comments. Indeed, there are many review articles devoted to the role of K+ in the cell activity, but they are concentrated around K+ transport systems (mainly K+ channels), their structure, properties, regulation in target strategies (as targets). The originality of the proposed review lies in the fact that it discusses K+ involvement in the regulation of proliferation and apoptosis. We focus on K+ content and transmembrane K+ fluxes in cultured animal cells to show the relationship between K+ transport and cellular growth, and discuss how K+ may contribute to fundamental changes of cell state leading from quiescent state to proliferation, senescence or apoptosis. Based on our own experimental data and theoretical analysis of ion homeostasis of animal cells we suggest that K+ may be important for cell proliferation as the main intracellular ion that is involved in cell volume regulation during cell transit from quiescence to proliferation, and came to conclusion that in dividing cells, the water content per cell protein should be higher than that in quiescent or differentiated cells. Cell hydration can significantly impact the biochemical life of the cell. Variations in the degree of hydration may alter the crowding of macromolecules within the cell thereby influencing the many regulatory events such as intracellular signaling, protein transport, and epigenetic modifications.

This review article underlines a link between intracellular K+ and cell proliferation and growth, which is mediated through regulation of cell volume and leads to a constancy of cellular K concentration and higher hydration of a proliferating cell as compared to a resting cell..

In accordance with your comments, improvements have been made in all sections of the review.

  1. There are 5 figures containing the summary data of studies on human mesenchymal stem cells (sections 2 and 3), human lymphocytes (section 4) and cells of permanent lines in culture (section 5).
  2. The Conclusion (Section 6.Concluding Remarks) has been revised.

Reviewer 2 Report

Comments and Suggestions for Authors

Ions play an important role in cell growth. The author review explores the role of K+ in maintaining cellular homeostasis during the transition of cells from quiescence to proliferation and the induction of apoptosis. As far as the topic is concerned, it is more meaningful. The author also chose some suitable topics to review and the content is also substantial. Therefore, I recommended this work can be accepted after some corrections need to be made listed as follows:

1.For the keywords, the author should choose more suitable keywords to increase the search.

2.The authors selected three types of cells, transformed cells in culture as well as human quiescent T cells and mesenchymal stem cells, why does the author focus on these three types of cells?

3.The author has carried out a lot of analysis in the article, but there are few graphs. It is suggested to add some pictures or tables, so that readers can understand it more easily.

4.For references, references are old, and authors should try to find newer ones.

5. The manuscript has some typographical and grammatical errors which should be corrected.

Author Response

Thank you for your comments, we have made the following corrections:

  1. New keywords added
  2. Cells were selected for the study according to the degree of their transformation and intensity of proliferation, since it is widely believed that transformed cells with a high rate of reproduction are characterized by increased ion fluxes through the membrane, a higher level of expression of ion transporters and an increased K+ content. It was on these cells that the dependence of K+ content on the growth rate of the culture was first shown and a connection with proliferation was revealed. Further, to analyze this relationship, normal human lymphocytes were used, on which basic data were obtained on the content of K+ and water during the transition of cells from a quiescent state to proliferation. Stem cells were used as a model of normal proliferating cells, which demonstrated ionic changes during the transition from proliferation to its arrest during culture aging and changes in the spatial culture system (2D and 3D configurations).
  3. There are 5 figures containing the summary data of our studies on human mesenchymal stem cells (sections 2 and 3), human lymphocytes (section 4) and cells of permanent lines (section 5).
  4. Indeed, there are many references to works of the 90s. It was at this time, in connection with the possibility of using cell cultures for studying ion-membrane processes, works appeared on the issue under consideration and the idea arose that ions are important for triggering proliferation and are involved in cell transformation. This is probably why works from the 90s are fully cited. Further, the question of the mechanism of participation of ions (K+, in particular) was considered mainly in connection with the participation of ion channels in triggering the proliferative response.

The new edition of the article has added more new links.

  1. Errors, grammatical and typographical, corrected. English text has been edited.

Reviewer 3 Report

Comments and Suggestions for Authors

Title: The role of intracellular potassium in cell quiescence, proliferation and death

Manuscript ID: ijms-2760359

I would like to inform you that review has been completed.  

This is an interesting review that explores the role of K+ in maintaining cellular homeostasis during the transition of cells from quiescence to proliferation and the induction of apoptosis. The authors focus on the relationship between intracellular K+, growth and proliferation rate of different cells including transformed cells in culture as well as human quiescent T cells and mesenchymal stem cells, and analyzed the concomitant changes in K+ and water content in both proliferating and apoptotic cells. The authors reported that during the initiation of cell proliferation and apoptosis changes in the K+ content in cells occur in parallel with changes in water content and therefore do not lead to significant changes in the intracellular K+ concentration. The authors concluded that K+ is involved in the regulation of cell volume during the transition from quiescence to proliferation and can be considered as a hallmark of cell proliferation and transformation.

Recommendation: Minor revision

1.    Sufficient information about the previous study findings should be presented for readers to follow the present review rationale and aims.  

2.    Illustrative figures should be added to properly reflect the relevant roles and mechanisms.

Comments on the Quality of English Language
  • Several sentences are confusing and sometimes misleading. There are places where the wording is somewhat awkward. I would suggest using an English Language Editing Service to ensure that your work is written in correct scientific English. Also, I would suggest more proof-reading and editing, ideally by a native speaker.
  •  

Author Response

Thank you for your comments, we have made the following corrections:

  1. In revised version, Section 1. Introduction contains links to previous studies that concern the involvement of monovalent ions in cell proliferation.
  2. 5 new figures are presented that contain data from our studies on human mesenchymal stem cells (Sections 2 and 3), human lymphocytes (Section 4) and cells of permanent lines (section 5).

Reviewer 4 Report

Comments and Suggestions for Authors

This manuscript is rather a brief review on the role of intracellular K+ in cell proliferation, differentiation, and degeneration. As the predominant monovalent cation in cells, K+ has been considered a house-keeping ion, but its specific signaling role has not been much explored in the literature. In this respect, a review summarizing what is known on this topic would be of keen interest to the science community. However, it seems that the manuscript does not convincingly deliver its messages with high impacts. It could be because the existing knowledge is not much in depth and width, but the manuscript appears to lack a novel perspective stemming from what has been summarized in the text. It often reads like a mere collection of facts and observations that do not point toward an interesting hypothesis. To fix the problem, the authors may want to consider serious restructuring of the text with extensive revisions and elaboration on figures, tables, and highlights.

Specific comments: 

11. The manuscript addresses complex changes in intracellular ion concentrations, cell volumes, and other factors that take place in cells undergoing proliferation, differentiation, cancer, apoptosis, etc.  However, there is not a single figure in this manuscript, which makes it very difficult for the readers to follow the point.  The text itself is often redundant where it repeats the same concept in several different places.  It will be useful if the authors provide “comprehensive figures” showing how K+ concentration (per protein or per volume) changes as a cell undergoes proliferation, differentiation, apoptosis, etc. Related to that, the provided Table is difficult to understand because the columns of numbers do not have clear specification. This has to be done more precisely.

22. The manuscript very often uses the expression like “K+ change is accompanied by…” which does not say much about the causality of the two changes. In this way, it would be hard to know what the real role of K+ is in the processes. Thus, it must be specified, whenever possible, which causes which, and how it is achieved. In other words, this manuscript does not provide mechanistic explanations in most of cases although it mentioned briefly on some enzymes and ion channels such as WNK kinase, Na/K-ATPase, etc. The manuscript must include much more discussions like that in order to explain what is really going on when K+ changes. To effectively conceptualize the point, the authors should provide also here some “comprehensive figures” with arrows, names, and numbers to make the story reasonable to follow. What is known can be specified, and what is not clear or speculative should be indicated with a question mark in such a figure. Abandoning readers without any figurative summary makes this manuscript very ineffective, and the story goes nowhere.

T3. The text should be restructured in accordance with the figures provided. Moreover, it could be said that the “Concluding Remark” does not conclude much. It only raises questions. It is exceptionally long and introduces other new facts and concepts that were not discussed in the text. There are even new citations. In other words, it is not a conclusion or conclusive remark. The text in Concluding Remark has to be presented as a new section, and the real conclusion should be place at the end of the manuscript to summarize all the important take-home messages and its significance.

44. The manuscript in its current form does not deliver what is expected, and the readers do not find much remaining in their hands when the last page is turned. A review article must be more than a collection of reference list. We want to hear some unique idea based on new perspective that the cited observations allow us to conceive or promote. One thing the authors might want to consider may be the possible involvement of mechanotransduction that links K+ and cell volume to physical forces and biochemical changes. It was taken as a vague example, but the authors are encouraged to present the topic in some new light and to inspire a sense of leading direction.

55. As a minor point, the text should be improved with the correct use of comma, and by limiting the occurrence of long sentences. Awkward expression should be minimized, too. This applies to the entire manuscript, but I take the abstract as an example in case authors may want to consider. i) Line 10, add “intracellular” in front of K+ and delete “in maintaining cellular homeostasis”; ii) Line 19, finish the sentence at “…to proliferation”, then the last sentence would be (after rearrangement): “The content of K+ and water in dividing (not cyclic) cells is higher than in quiescent or differentiated cells, which can be considered as a hallmark of cell proliferation and transformation.  Nevertheless, this is a very dilute conclusion of an abstract or a paper in my opinion, and that is why the authors should restructure the manuscript and find their way to present and highlight their important points to the science community.                              

Comments on the Quality of English Language

Punctuation rules should be respected. Long sentences should be avoided. Syntax should be considered carefully so that the scientific contents would not contradict each other in the same sentence. 

Author Response

Thank you for your comments, we have made the following corrections:

  1. English text has been edited.

The manuscript has been revised. 5 new figures are given that contain summary data from studies establishing a connection between changes in intracellular K+ and the initiation of proliferation and apoptosis.

In the revised version, the review first states the phenomenon of a decrease in K+ content (normalized to the mass of cellular protein) during the growth of cultures of transformed cells of permanent lines (Section 2, Table 1). It further confirms the identified phenomenon in studies on actively proliferating human mesenchymal stem cells (Fig.1) and non-proliferating, stopped as a result of oxidative stress (senescence) (Fig. 2). Cells were selected for the study according to the degree of their transformation and intensity of proliferation, since it is widely believed that transformed cells with a high rate of reproduction are characterized by increased ion fluxes through the membrane, a higher level of expression of ion transporters and an increased K+ content. It was on these cells that the dependence of K+ content on the growth rate of the culture was first shown and a connection with proliferation was revealed. 

Section 4 (Fig. 3 and 4) provides an analysis of the relationship between the level of intracellular K+  and proliferation, namely: 1) an increase in the intracellular K+ content is revealed during the transition of activated T lymphocytes from a state of rest to proliferation, 2) a new fact is established of an increase in the content of cellular water in the process of activation of human T lymphocytes, 3) an increased water content in leukemia cells is detected, 4) a proportionality is found between the increase in water content and K+ content in cells in the process of activation of T lymphocytes, and 5) a conclusion is made about the constancy of the K+ concentration in cells when they change proliferative status.

Section 5 (Fig. 5) analyzes the content of K+ and Na+ and K+ influxes, as well as changes in water content when apoptosis is triggered. Edits have been made in the Table.

  1. In the revised manuscript, following the new summary figures, we establish the relationship between intracellular K+ and cellular growth, which is mediated by the regulation of cell volume. Based on these experimental data and theoretical analysis of ion homeostasis of animal cells we suggested that K+ may be important for cell proliferation as the main intracellular ion that is involved in cell volume regulation during transit from quiescence to proliferation, and came to the conclusion that in dividing cells, the water content per cell protein (i.e., the degree of cell hydration) should be higher than in quiescent or differentiated cells. And in fact, the water content in cancer cells is higher than in normal cells of similar origin, while age-related loss of intracellular water and cell dehydration are concomitant factors of cellular aging. We further speculate that cell hydration can significantly impact the biochemical life of the cell. Variations in the degree of hydration may alter the crowding of macromolecules within the cell thereby influencing the many regulatory events such as intracellular signaling, protein transport, and epigenetic modifications.

  1. The text has been reconstructed in accordance with new 5 drawings. A detailed description of the corrections made is given in the response to comment 1.

Conclusion (Section 6. Concluding Remarks) has been revised.

  1. The novelty of the proposed review is as follows. When discussing the mechanisms of regulation of cell proliferation, researchers first analyze chains of interconnected “fast” events that perform a signaling function and control the triggering of specific cellular functions of the cell. Monovalent ions, through changes in their fluxes, intracellular concentrations, and membrane potential, are involved in a chain of receptor-mediated events when triggering proliferative, apoptotic and other cellular responses. These rapid ionic changes involve membrane ion transport systems (pumps, ion channels and coupled ion transporters) and are being intensively studied. The main reviews concerned the role of ions in proliferation are specifically devoted to fast “signaling” ion events and their direct or indirect role in cell activation processes.

We draw attention to the fact that K+ is the most abundant monovalent ion in the cell and therefore can hardly perform a signaling function. To clarify the mechanism by which K+ may be involved in the regulation of proliferation, we focused on measurements of intracellular K+ content and transmembrane K+ fluxes in cultured animal cells and came to the conclusion that a link between intracellular K+ and cell proliferation is mediated through regulation of cell volume leading to a constant of cellular K+ concentration. The participation of K+ in the regulation of the volume of proliferating or apoptotic cells has been considered previously (Lang et al., 1995,1997). Mechantoransduction that links ion movement (Na channels, for example), cell volume and cell activity is also an important aspect of the problem (Morachevskaya, Sudarikova, 2021). We are discussing the mechanism by which K+ may contribute to fundamental changes in cell state leading from quiescent state to proliferation, senescence or apoptosis and some unique idea is as follows. K+ may be important for cell proliferation as the main intracellular ion that is involved in cell volume regulation during cell transit from quiescence to proliferation, and came to conclusion that in dividing cells, the water content per cell protein should be higher than that in quiescent and differentiated cells. A change in the degree of hydration of dividing cells compared to resting cells can impact the biochemical life of the cell.

This review highlights the relationship between intracellular K+ and cell proliferation and growth, which may be mediated by cell volume regulation and results in constant cellular K+ concentrations. The reviews offered for citation (Ref. 16-19) concern the functioning of K+ channels in connection with their participation in proliferation and apoptosis.

  1. Extensive editing of English language is done throughout the text.

Manuscript has been edited according to comments.

Detailed comments on the edited text of the review are given above, in response to remark 1. (see above   … “5 figures are given that contain the summary data of our studies of the connection between…….”.    

Lines 10 and 19 are corrected.

The Conclusion (Section 6. Concluding Remarks) has been revised.

Round 2

Reviewer 1 Report

Comments and Suggestions for Authors

The authors have done great efforts in improving the manuscript. The revised manuscript can be accepted.

Comments on the Quality of English Language

Minor editing.